# Assessing the quality of tuberculosis care using routine surveillance data: a process evaluation employing the Zero TB Indicator Framework in Mongolia

Ariunzaya Saranjav [1], Christina Parisi,[2] Xin Zhou,[3,4] Khulan Dorjnamjil,[1] Tumurkhuyag Samdan,[1,5] Sumiya Erdenebaatar,[1] Altantogoskhon Chuluun,[6] Tserendagva Dalkh,[7] Gantungalag Ganbaatar,[8] Meredith B Brooks [9], Donna Spiegelman,[4] Davaasambuu Ganmaa,[10] J Lucian Davis [2,11]

AS and CP contributed equally. DG and JLD contributed equally.

For numbered affiliations see end of article.

**Correspondence to**
Dr Ariunzaya Saranjav;
ariunzaya.sar@gmail.com

## ABSTRACT

**Objectives** To evaluate the feasibility of the Zero TB Indicator Framework as a tool for assessing the quality of tuberculosis (TB) case-finding, treatment and prevention services in Mongolia.

**Setting** Primary health centres, TB dispensaries, and surrounding communities in four districts of Mongolia.

**Design** Three retrospective cross-sectional cohort studies, and two longitudinal studies each individually nested in one of the cohort studies.

**Participants** 15 947 community members from high TB-risk populations; 8518 patients screened for TB in primary health centres and referred to dispensaries; 857 patients with index TB and 2352 household contacts.

**Primary and secondary outcome measures** 14 indicators of the quality of TB care defined by the Zero TB Indicator Framework and organised into three care cascades, evaluating community-based active case-finding, passive case-finding in health facilities and TB screening and prevention among close contacts; individual and health-system predictors of these indicators.

**Results** The cumulative proportions of participants receiving guideline-adherent care varied widely, from 96% for community-based active case-finding, to 79% for TB preventive therapy among household contacts, to only 67% for passive case-finding in primary health centres and TB dispensaries (range: 29%–80% across districts). The odds of patients completing active TB treatment decreased substantially with increasing age (aOR: 0.76 per decade, 95% CI: 0.71 to 0.83, p<0.001) and among men (aOR: 0.56, 95% CI: 0.36 to 0.88, p=0.013). Contacts of older index patients also had lower odds of initiating and completing of TB preventive therapy (aOR: 0.60 per decade, 95% CI: 0.38 to 0.93, p=0.022).

**Conclusions** The Zero TB Framework provided a feasible and adaptable approach for using routine surveillance data to evaluate the quality of TB care and identify associated individual and health system factors. Future research should evaluate strategies for collecting process indicators more efficiently; gather qualitative data on explanations for low-quality care; and deploy quality improvement interventions.

## STRENGTHS AND LIMITATIONS OF THIS STUDY

⇒ This study provides a comprehensive evaluation of the quality of tuberculosis (TB) diagnostic, treatment and prevention services in Mongolia using routine surveillance data.

⇒ This study provides a method for identifying individual and health-system factors associated with lower quality care, which may facilitate targeting of quality improvement interventions.

⇒ Routine TB surveillance data are susceptible to outcome misclassification in the absence of unique identifiers.

⇒ Results may not be generalisable to all TB units in Mongolia given the purposive sampling strategy.

## INTRODUCTION

Tuberculosis (TB) incidence and mortality rates have been declining worldwide for two decades, but with recent global disruptions to TB services and surging poverty rates related to the COVID-19 pandemic, the absolute number of annual TB cases (10 million) and deaths (1.4 million) is likely to increase significantly in the coming years.[1] As the global TB community prepares for surging caseloads and declining resources, the crisis in the quality of TB care has never been more salient. Before the pandemic, one-third of patients with TB did not access care or were not reported to public health authorities, and even among those who sought care for TB, there was wide geographic variability in treatment coverage and success rates.[2] Other evidence-based strategies for reducing TB incidence, including active case-finding[3 4] and TB preventive therapy (TPT)[5] remain unavailable in most low-income and middle-income countries.[1] The Lancet Commission on High Quality Health Systems has estimated that

almost 50% of annual TB deaths may be attributable to low-quality care.[6]

Although improving access to high-quality services is the ultimate goal, there is first a need for a revolution in measuring and reporting on quality. The general metrics of high-quality care—safety, timeliness, effectiveness, efficiency, equity, patient-centeredness and integration of services—are widely agreed upon.[7 8] However, these metrics have not been widely adapted or adopted for reporting data from TB programmes in real-time. For example, National TB programmes (NTPs) have traditionally focused public health reporting on treatment coverage and treatment success rates, two narrow measures of effectiveness, and only recently have added process measures such as the proportion of eligible patients initiating TPT.[9–11] If improving quality is to be 'hard-wired' into NTPs, as a recent Lancet Commission on Tuberculosis recommended,[12] there is an urgent need for research on expanding measures of quality for use as benchmarks, as called for by the WHO TB Programme[13] and others,[14] starting with measures that make better use of routine TB surveillance data.[15]

Structured evaluations of care processes through cascade analyses have increasingly been proposed[14 16–21] and implemented to describe gaps in quality of TB care in low-income and middle-income countries, including for clinic-based TB diagnosis and treatment in adults[22–31] and children,[32 33] active case-finding[28 34 35] and prevention[28 36 37] (online supplemental appendix S1). The Zero TB Initiative[38] has introduced an expansive quality indicator framework that integrates all three of these cascade types[18]; similar approaches have been proposed for persons living with HIV.[17] Here, we evaluate the feasibility of the Zero TB Indicator Framework as a tool for assessing the quality of TB services, using data from Mongolia, a Zero TB Initiative country.

## METHODS
### Setting
Mongolia, a lower middle-income country of 3 million people, has the fourth-highest TB incidence in the WHO Western Pacific Region, at 428 cases per 100 000, and among the world's lowest TB treatment coverage rates, at 31%.[1] Mongolia's high TB prevalence rate (757 per 100 000) exceeds its annual TB incidence rate, indicating a prolonged average duration of illness and the presence of diagnostic delays before treatment.[11 39] More favourably, Mongolia's TB treatment success rate of 91% surpasses the global average, and the country funds 77% of its TB programme from domestic sources.[1]

The Mongolian Government provides free TB services in centralised TB dispensaries located in every urban district and rural province. NTP guidelines recommend TB symptom screening of all individuals attending primary health centres and referral of those screening positive to centralised dispensaries for diagnostic evaluation with sputum smear microscopy and chest radiography.

Molecular testing is available only in select dispensaries and the National Center for Communicable Diseases (NCCD), where mycobacterial culture is also available. NTP guidelines recommend active TB case-finding among high-risk groups (people living with HIV; miners; those unhoused, incarcerated or living in poverty; people living in remote areas; healthcare workers; school staff; orphans) using community-based symptom screening and referral to dispensaries for chest radiography. NTP guidelines also recommend referral of all household contacts of patients diagnosed with sputum smear-positive TB to dispensaries for TB symptom screening and chest radiography, plus tuberculin skin testing (TST); TST-positive children without evidence of active TB are eligible for TPT with 6 months of daily isoniazid.

The NTP oversees management of the paper registries used for TB reporting, as well as the entry of data from these source documents into the national electronic TB information database. NTP compiles surveillance data quarterly and provides technical oversight and assistance to TB providers in Mongolia's 21 provinces and 9 capital city districts. Since 2017, the NTP has implemented enhanced human resource management practices for health workers in TB dispensaries, including on-the-job training in TB care, performance incentives, and retirement benefits.

### Study design, participants and sites
Between 1 January and 31 December 2017, we retrospectively performed three retrospective cross-sectional cohort studies, and two longitudinal studies each nested in one of the cohort studies, to construct the three Zero TB Indicator Framework cascades. For the Search, Treat, and Prevent cascades, we conducted retrospective cross-sectional cohort studies. For the Treat and Prevent cascades, we had access to individual data, so we were able to conduct nested longitudinal cohort studies to examine long-term outcomes of treatment in addition to the baseline cross-sectional cohort studies. The Search cascade examined community-based, active TB case-finding carried out among selected high-risk populations in the capital city districts by the NCCD and its partner non-governmental organisations (NGOs), including the Korean National Tuberculosis Association and the Mongolian Health Initiative. Screening targeted people living in low-income census tracts (identified using data from the Mongolian National Statistics Office), as well as staff and students at a school with a recent TB outbreak. The Treat cascade examined clinic-based TB diagnosis and treatment among patients with possible TB identified in primary health centres and referred for evaluation at TB dispensaries, including first episodes of care only. The indications for referral followed NTP guidelines and included reporting one or more TB symptoms, defined as a cough and/or fever lasting ≥2 weeks, weight loss or blood-tinged sputum; or being found to have abnormalities on chest radiography. Finally, the Prevent cascade examined diagnosis of latent TB infection (LTBI) and

delivery of TPT among household contacts of patients with index TB .

Bayanzurkh (population 280,000), Khan-Uul (population 170,000), and Chingeltei (population 150,000) are three densely populated districts among nine total districts in the Mongolian capital, Ulaanbaatar, all of which have both urban and rural areas. Mandal is the highest-population sub-province in rural Selenge province. Case notification rates in all four communities approximate the national average case notification rate of 124/100,000 persons. We chose Bayanzurkh District because it has the highest TB rate in Ulaanbaatar; Khan-Uul District because it is a partner district for the Mongolia Zero TB Initiative and has established active surveillance and screening programs at secondary schools; Chingeltei District because it was recognized as a model district at the 2018 National TB Forum; and Mandal Sub-Province because it has two prisons and three large mines, all of which are high-risk settings for TB. Each community has a dedicated dispensary for TB care: Bayanzurkh has 15 staff members (7 TB specialist doctors, 6 TB nurses, 2 lab technicians); Khan-Uul, 7 staff members (4 TB specialist doctors, 2 TB nurses, 1 lab technician); Chingeltei, 5 staff members (2 specialist TB doctors, 2 nurses, 1 lab technician); and Mandal, 7 staff members (1 TB doctor, 5 TB nurses, 1 lab technician). Each dispensary also has two to four volunteers.

## Measurements

Mongolian Health Initiative staff (SA, DK, ST, ES) digitally photographed official NTP laboratory, treatment and contact registers during site visits. Using these photos, they double-entered demographic and clinical data on patients with possible TB, patients with confirmed TB and household TB contacts into electronic data collection forms (REDCap, Nashville, Tennessee, USA).They also extracted aggregated data on active case-finding from NGO reports to the NTP, and on TB symptom screening in primary health centres from records of the Mongolian Association of Family Medicine Specialists.

We successfully defined and adapted 14 of 16 Zero TB Indicators to fit data routinely collected by the Mongolia NTP and partners (table 1),[18] and proposed evidence-based performance targets derived from published guidelines or systematic reviews (online supplemental appendix S2). We could not obtain data on which individuals were TB-free 1 year after completing active TB treatment or TPT because post-treatment follow-up is not routinely performed in Mongolia. Patients with TB treatment failure are managed according to national guidelines, which in 2017 recommended repeat sputum examination for mycobacterial culture and drug-susceptibility testing, and empiric initiation of a WHO Category II re-treatment regimen.

## Analysis

We presented baseline demographic and clinical characteristics of those completing referral for TB evaluation for the Treat and Prevent studies only; demographic and clinical characteristics were not available for the Search study. We calculated proportions for dichotomous variables and medians with upper and lower quartiles for continuous variables. For t-tests and $\chi^2$ tests for significance, a p value<0.05 was considered significant.

For the process evaluation, we determined indicators for the Search, Treat and Prevent cascades differently. The Search and Treat cascade indicators were calculated as simple proportions, and the yield as counts diagnosed and initiating treatment divided by total counts screened, with the number-needed-to-screen reported as the inverse of this ratio. Within the Prevent Cascade, we assessed for correlation of process outcomes among contacts within households by calculating intraclass correlation coefficients (ICCs). For outcomes with ICC ≥0.10, we estimated the probability of completion by fitting a multivariate, logistic regression model using generalised estimating equations (GEE)[40] with an exchangeable working correlation structure to account for households. For each cascade, we presented indicators within flow diagrams (figure 1) and calculated stepwise and cumulative probabilities of receiving guideline-adherent care. For the Treat Cascade, we also calculated site-level cumulative probabilities of symptomatic individuals at primary health centres being referred to a dispensary, evaluated for active TB, and starting and completing treatment if diagnosed. We derived a formula to estimate 95% CIs for cumulative probabilities (online supplemental appendix S3). We assessed between-site differences in performance using the $\chi^2$ test of independence. Finally, for process outcomes with the greatest losses (ie, the largest percentage drops in retention between steps), we constructed multivariate, logistic regression models using GEE to identify individual demographic (age, gender) predictors of dropout, with 95% CI and p values provided by Wald tests. We did this for the 'starting active TB treatment' and 'completing active TB treatment' steps in the Treat cascade (see online supplemental tables S1 and S2) and the 'completing TB evaluation' and 'being prescribed and initiating TPT' steps in the Prevent cascade (see online supplemental tables S3 and S4). We did not do this for the other steps because completion rates were high and the numbers who were lost were not sufficient to power a predictive model. We based sample size on available participants. We performed all analyses in SAS (V.9.4, The SAS Institute, Cary, North Carolina, USA).

## Patient and public involvement in research

Patients who concurrently or previously had TB were not involved in defining the research design or implementation, but members of the research team (AS and KD) met intermittently with staff at the primary health centres and the dispensaries during data collection to refine the study measures and to disseminate the research findings.

**Table 1** Process indicators of performance, adapted for Mongolia from the Zero TB Indicator Framework

| Indicator (proportion) | Numerator (count) | Denominator (count) | Interpretation | Target/source |
|---|---|---|---|---|
| **Search Cascade*** | | | | |
| 1. (Screening) Coverage | Screened for active TB disease† | Estimated individuals in target population | Effectiveness of outreach to target population | 100%[62] |
| 2. Positive TB symptom screen | Positive TB symptoms or CXR screening | Screened for active TB disease | Accuracy and efficiency of implementation of screening | 25%–50%[62] |
| 3. Diagnostic evaluation | Completed TB evaluation‡ | Positive TB symptoms or CXR screening | Effectiveness of linkage to testing | 100%[62] |
| 4. (Active) TB Diagnosis | Diagnosed with active TB disease§ | Completed TB evaluation | Yield of diagnostic evaluation | 10%–20%[63] |
| 5. Linkage to TB Treatment | Prescribed active TB treatment¶ | Diagnosed with active TB disease | Effectiveness of linkage to treatment | 100%[64] |
| **Treat Cascade** | | | | |
| 1. (Active) TB Diagnosis | Diagnosed with active TB disease§ | Evaluated for active TB disease | Yield of diagnostic evaluation | 10%–20%[63] |
| 2. Bacteriologic Confirmation | MTB-positive bacteriologic test*** | Diagnosed with active TB disease | Accuracy and efficiency of implementation of screening | 70%–90%[65 66] |
| 3. Linkage to TB Treatment | Prescribed active TB treatment¶ | Diagnosed with active TB disease | Effectiveness of linkage to treatment | 100%[64] |
| 4. Treatment Success | Treatment outcome successful¶†† | Started active TB treatment | Effectiveness of treatment | ≥90%[10] |
| 5. TB-free Survival | TB-free 1 year later | Treatment outcome successful | Effectiveness of treatment follow-up | ≥95%[54] |
| **Prevent Cascade** | | | | |
| 1. TB Screening | Completed TB screening‡‡ | Estimated household contacts of patients diagnosed with smear-positive TB | Effectiveness of outreach to target population | 100%[67] |
| 2. TB Evaluation | Completed TB clinical/lab evaluation§§ | Completed TB screening | Effectiveness of linkage to testing | ≥90%[10] |
| 3. (Active) TB Diagnosis | Diagnosed with active TB disease§ | Completed TB clinical/lab evaluation | Yield of diagnostic evaluation | 10%–20%[63] |
| 4. TPT Eligibility | Confirmed not to have active TB disease & TST-positive§§ | Completed TB clinical/lab evaluation | Accuracy and efficiency of implementation of testing | 95%[5] |
| 5. (TPT) Prescription | Prescribed preventive therapy¶¶ | Eligible for preventive therapy | Adoption of TPT by providers | ≥90%[10] |
| 6. (TPT) Uptake | Started preventive therapy | Prescribed preventive therapy | Reach of TPT to patients | 100%[5] |
| 7. (TPT) Completion | Completed preventive therapy | Started preventive therapy | Effectiveness of TPT implementation | ≥80%[5] |
| 8. TB-free survival | TB-free 1 year later | No TB present at initial evaluation | Effectiveness of treatment follow-up | 100%[5] |

Indicators, cascades and numbering derived from the Zero TB Indicator Framework,[1] with numerators adapted using the Mongolian National TB Guidelines and source data, and targets derived from WHO guidelines and targets. Additional details available in online supplemental appendix S2.

*The target population included people living in poverty, school staff and students at a school with a recent TB outbreak, and household contacts of patients with index TB.

†Screening includes questions about a history of TB symptoms (cough or fever lasting ≥2 weeks, weight loss or bloody sputum) and/or CXR, with any symptom or CXR abnormality (eg, consolidation, infiltrates, nodules, cavities) defining a positive screen.

‡Completing TB evaluation is defined as receiving a clinical assessment and all tests required by the TB clinician at the TB dispensary. According to NTP guidelines, these may include direct (sputum smear microscopy, mycobacterial culture or Xpert MTB/RIF assay) or indirect (biopsy, ADA, TST) tests.

§Diagnosed with active TB disease is defined as being recorded in the National TB Programme TB lab register and/or the treatment register as a new patient with TB, confirmed as one of the following categories: bacteriologically confirmed drug-susceptible TB, bacteriologically confirmed drug-resistant TB, bacteriologically confirmed multidrug-resistant TB, bacteriologically confirmed extensively drug-resistant TB or unconfirmed TB based on clinician judgement.

¶Defined as being recorded in the National TB Programme treatment register as a new patient with TB prescribed TB treatment.

**Bacteriologic confirmation may occur by sputum smear microscopy, sputum mycobacterial culture and/or Xpert MTB/RIF.

††Treatment outcome successful was defined to include all patients who completed the recommended duration of treatment, whether documented to have negative smear microscopy or a negative mycobacterial culture result within the final month of treatment ('TB cured', per NTP guidelines) or not ('Completed TB treatment', per NTP guidelines). Patients lost to follow-up (ie, those who did not start treatment or had treatment interrupted for ≥2 consecutive months); those not evaluated; and those who transferred to another treatment unit in a different area, were defined as treatment outcome not successful.

‡‡Those eligible for screening for possible preventive TB treatment include all household and other close contacts of a patient with index TB. Screening should be performed within 14 days following the diagnosis of an patient with index TB. Screening may be repeated 12 weeks after exposure.

§§According to NTP guidelines, contacts 0–15 years of age should be evaluated with a TST (defined as positive for TB infection among close contacts when ≥5 mm skin induration is documented within 2–3 days of placement); TST may be repeated 12 weeks after exposure. For contacts ≥15 years, a CXR is required. If TST is positive or CXR abnormalities are identified, bacteriological testing with sputum Xpert MTB/RIF or mycobacterial culture is recommended. If TB is bacteriologically confirmed, active TB treatment is recommended. For children under 5 years of age who are TST positive and in whom active TB has been excluded, TPT is recommended, and may be prescribed in other situations (eg, concerns about false-negative results) at the clinician's discretion.

¶¶Preventive therapy for children is 10 mg/kg of isoniazid for 6 months. The daily maximum dose of isoniazid is 300 mg.

ADA, adenosine deaminase; CXR, chest radiography; NTP, national tuberculosis programme; TB, tuberculosis; TPT, TB preventive therapy; TST, tuberculin skin test.

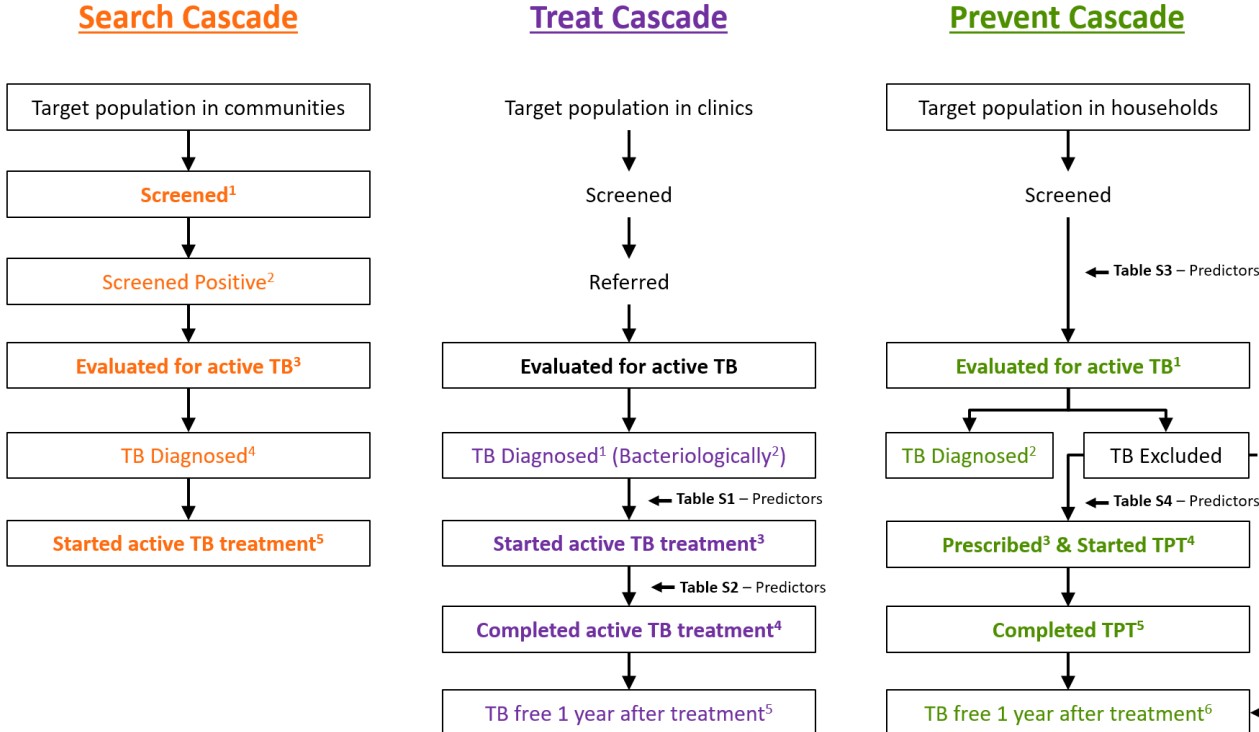

**Figure 1** Zero TB Indicator Framework Cascades, adapted for Mongolia. Flow diagrams showing the movement of individuals through each individual step of the three TB delivery cascades within the Zero TB Indicator Framework. These have been modified to fit data available in Mongolia, and colour coded to differentiate among the Search, Treat and Prevent service cascades. Steps framed within a rectangular box are derived from the Zero TB Indicator Framework, while unframed steps indicate additional important steps within the cascades. Solid arrows show the direction of patient flow through the cascades; a dashed arrow shows where excluded individuals re-enter the cascade. Coloured box labels correspond to the 16 Zero TB Framework indicators, with superscripts identifying indicator numbers as defined in table 1; five come from the Search Cascade, five from the Treat Cascade and six from the Prevent Cascade. The indicators in **bolded text** describe the quality of care at a given step in the cascade, while the non-bolded indicators describe the diagnostic and/or epidemiological yield of that step in TB care. Finally, brief references to the following supplemental tables are located adjacent to the steps where they will identify barriers to TB care through multivariate models: online supplemental table S1, Predictors of Starting Active TB treatment (Treat Cascade); online supplemental table S2, Predictors of Completing Active TB Treatment (Treat Cascade); online supplemental table S3, Predictors of Completing TB Evaluation (Prevent Cascade); online supplemental table S4, Predictors of Being Prescribed and Initiating TPT (Prevent Cascade). TB, tuberculosis.

## RESULTS

### Search cascade

There were 15 947 people in high-risk populations screened for active TB within the four study communities, including 6670 (42%) in Chingeltei district, 4632 (29%) in Bayanzurkh district, 3865 (24%) in Khan-Uul district and 780 (5%) in Mandal subprovince. This included 12 082 (76%) vulnerable individuals of low socioeconomic status in Chingeltei and Bayanzurkh districts and in Mandal subprovince; and 3865 (24%) students, teachers and administrative officials in Khan-Uul district. Of the original 15 947, a total of 15 294 (96%) underwent symptom screening and chest radiography for possible active TB disease (figure 2). Screening was positive for 1592 (10%). Of these, 1571 (99%) completed TB evaluation, with 97 (6%) diagnosed with active TB disease and 89 (92%) starting treatment.

Of all 15 947 community members screened for active TB in the Search Cascade, 15 261 (95.7%) completed each of the required steps of the cascade, an indication of high-quality care (figure 3A). The cumulative yield of active TB treatments among these communities was 558 per 100 000 screened, providing a number-needed-to-screen of 179.

### Treat cascade

There were 8518 people who screened positive for TB symptoms at 62 primary healthcare centres spread across the four communities (figure 4), including 3245 (38%) from Bayanzurkh, 2620 (31%) from Khan-Uul, 2160 (25%) from Chingeltei and 493 (6%) from Mandal. Of the 8518 individuals screened, 1486 (17%) were assigned non-TB diagnoses. The remaining 7032 (83%) were referred to dispensaries for TB evaluation, but only 4416 (63%) arrived and underwent and completed TB testing. Those tested included 2348 (53%) males and 4 people living with HIV (0.1%). Median age was 40 years (lower quartile (Q1) 26 to upper quartile (Q3) 55). Of

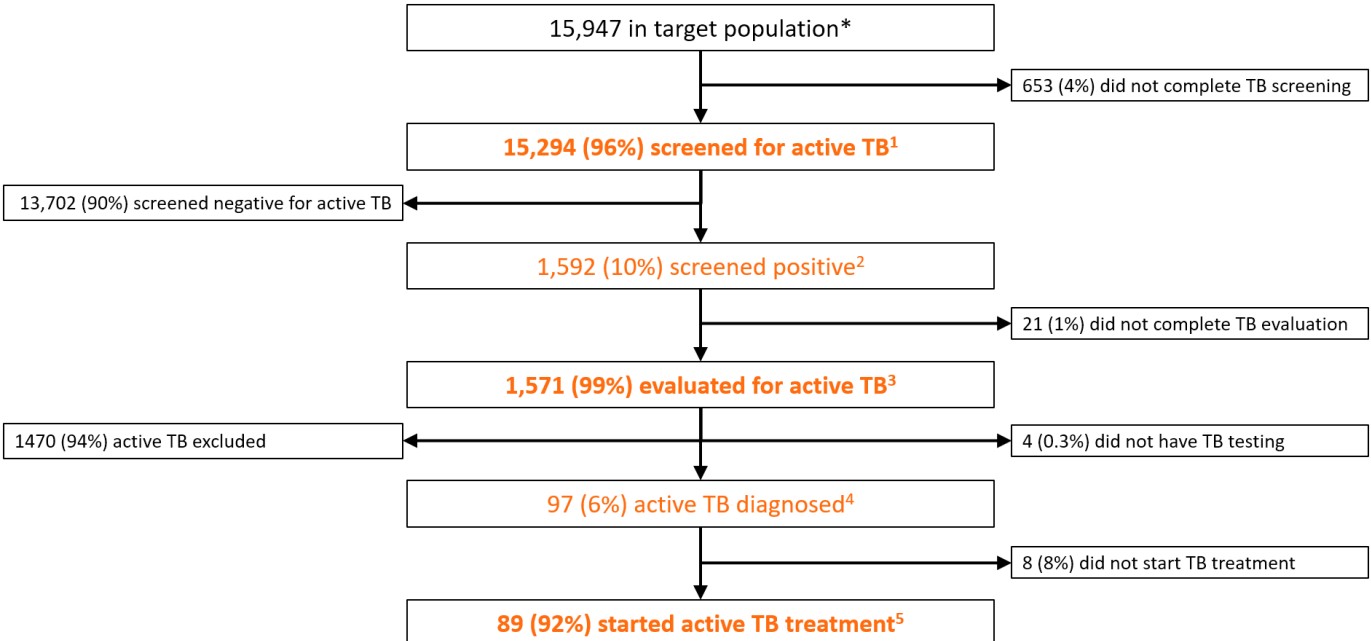

**Figure 2** Flow diagram showing the Search Cascade as adapted from the Zero TB Indicator Framework for Mongolia. Flow diagrams showing individuals entering and exiting at each step of the three cascades, either appropriately to the left side after having received guideline-adherent care or inappropriately to the right side after having received guideline-non-adherent care. Zero TB Indicators Treat indicators are shown in orange, with superscripts identifying indicator numbers as defined in table 1. The indicators in **bolded text** describe the quality of care at a given step in the cascade, while the non-bolded indicators describe the diagnostic and/or epidemiological yield of that step in TB care. TB, tuberculosis.

those visiting the dispensaries and tested, 1162 (26%) were diagnosed with active TB, 824 (71%) of whom were bacteriologically confirmed. Of these, 192 (23%) had drug-resistant TB. Among the 1124 (97%) who began treatment, 868 (77%) were successfully treated, including 81% of drug-susceptible and 55% of drug-resistant cases. Among those who did not complete treatment, 94 (8%) were lost to follow-up, 52 (5%) failed treatment, 31 (3%) died and 79 (7%) had unknown treatment outcomes.

Of all 8518 individuals entering the Treat Cascade by screening positive for TB symptoms at primary health centres, 5726 (67%) received all services defined as indicative of high-quality care in terms of timeliness and efficiency[7 8] (figure 3B).

Treat cascade indicators showed significant variation across sites (online supplemental table S0). Among those referred from primary health centres, evaluation rates ranged widely, from nearly 100% in Mandal *soum* to just 47% in Bayanzurkh ($\chi^2$ test for independence, p<0.001). The proportions starting treatment, in contrast, were consistently high, from 93% to 98% across sites (p=0.090). Finally, treatment success varied markedly, with a high of 87% in Mandal *soum* to a low of 72% in Bayanzurkh (p<0.001). The cumulative probabilities of being properly diagnosed and treated for TB given a positive TB diagnosis also varied substantially(p<0.001) and were highest at Mandal *soum* (80%, 95% CI: 68% to 92%) and lowest at Bayanzurkh (29%, 95%: CI 27% to 31%), with

similarly low probabilities at Khan-Uul (42%, 95% CI: 40% to 45%) and Chingeltei (44%, 95% CI 41% to 47%).

The odds of starting treatment among patients with active TB were lower among older individuals (adjusted OR (aOR): 0.72 per decade, 95% CI: 0.66 to 0.79, p<0.001) and males (aOR: 0.73, 95% CI: 0.62 to 0.86, p<0.001) (online supplemental table S1). The odds of completing TB treatment were also lower in older individuals (aOR: 0.76 per decade; 95% CI: 0.71 to 0.83, p<0.001), and males (aOR: 0.56; 95% CI: 0.36 to 0.88, p=0.013) (online supplemental table S2).

### Prevent cascade

Among 857 index patients diagnosed with smear-positive active TB, 75 lived alone or did not report on household contacts, leaving 782 (91%) eligible index patients with 2352 household contacts (mean 3 household contacts per household) eligible for contact investigation. Among contacts in the same household, there was modest correlation in eligibility for TPT (ICC=0.14), and a moderate-to-high correlation for evaluation completion (ICC=0.30) and prescription and initiation of TPT (ICC=0.41).

Of the 2352 household contacts in the target population, including 270 children under 5 years old, only 1932 (82%, 95% CI: 80% to 84%) visited the dispensary and were evaluated for active TB (figure 5).

Older patients were less likely to complete TB evaluation (aOR: 0.92 per decade, 95% CI: 0.86 to 0.98, p=0.015), after adjusting for the gender of the contact, the contact's

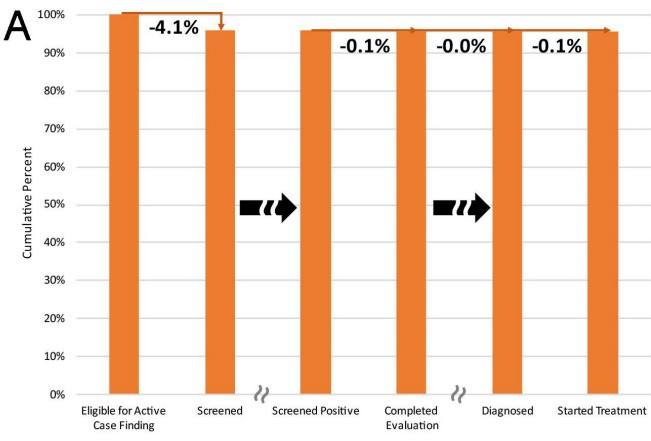

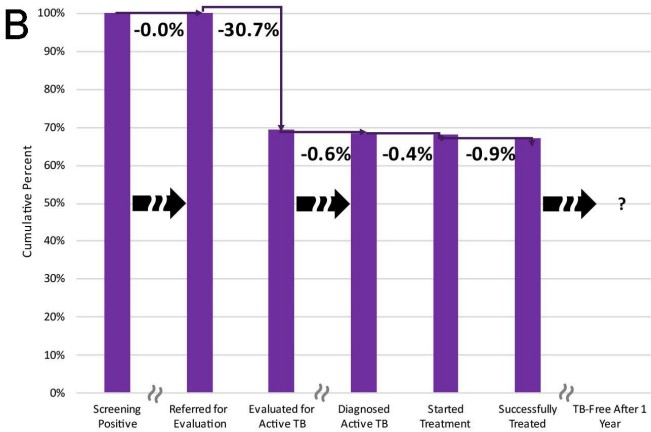

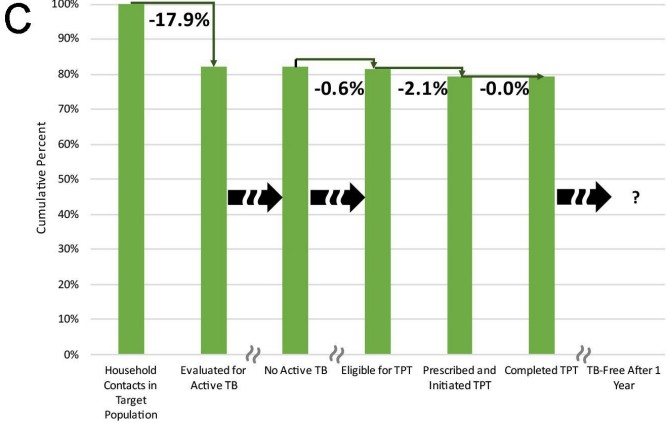

**Figure 3** Cumulative proportions of participants receiving high-quality TB care, by delivery cascade. Panel A: Search Cascade (n=15 947). Panel B: Treat Cascade (n=8518). Panel C: Prevent Cascade (n=2352). Bar graphs showing the cumulative probability of completing the key processes with each of the three TB delivery cascades. The probabilities for each step, shown as bars, were calculated as the simple proportion of all individuals receiving guideline-recommended care at the end of that step and all previous steps, divided by the total number entering the cascade. Thin elbow connector lines show the percentage lost with each step, calculated as the simple proportion of individuals not receiving guideline-recommended care at that step divided by the total number entering the cascade. Block arrows with discontinuity lines shows steps where individuals exit the cascade having received all guideline-recommended care. The question mark indicates that follow-up data on outcomes at 1 year were not available as these are not routinely collected in Mongolia. TB, tuberculosis.

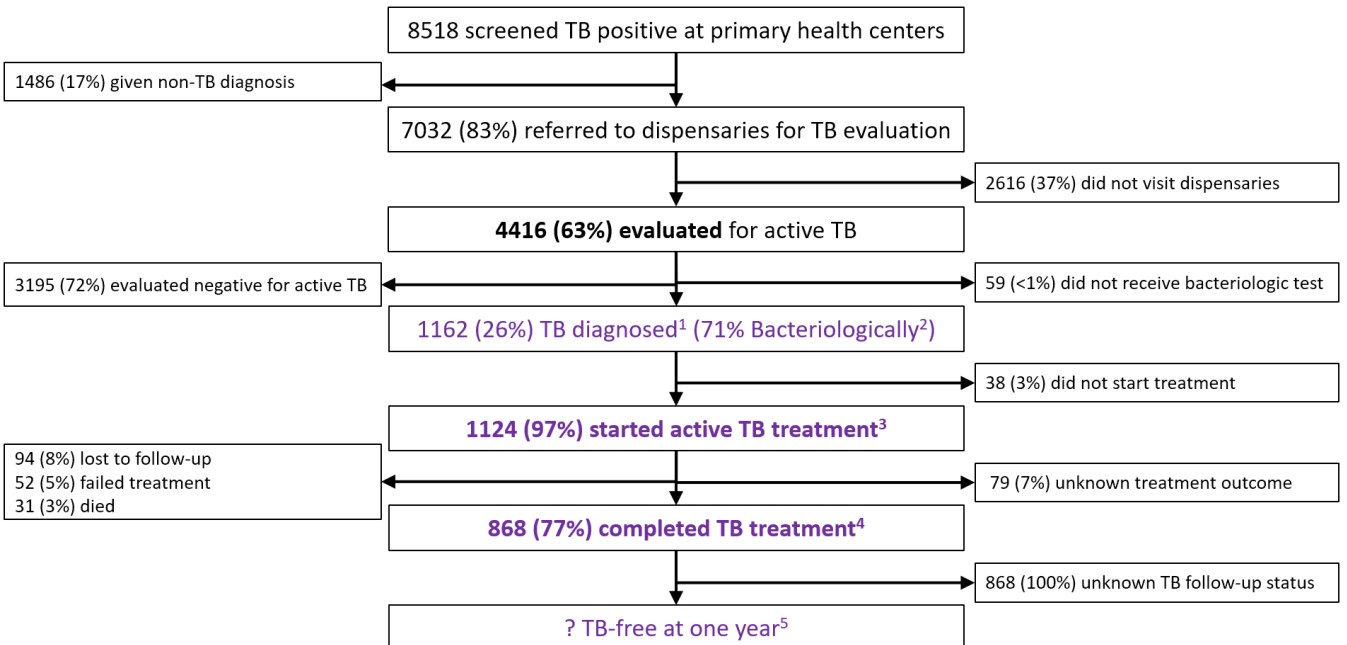

**Figure 4** Flow diagram showing the Treat Cascade as adapted from the Zero TB Indicator Framework for Mongolia. Flow diagrams showing individuals entering and exiting at each step of the three cascades, either appropriately to the left side after having received guideline-adherent care or inappropriately to the right side after having received guideline-non-adherent care. Zero TB Treat indicators are shown in purple, with superscripts identifying indicator numbers as defined in table 1. The indicators in **bolded text** describe the quality of care at a given step in the cascade, while the non-bolded indicators describe the diagnostic and/or epidemiological yield of that step in TB care. TB, tuberculosis.

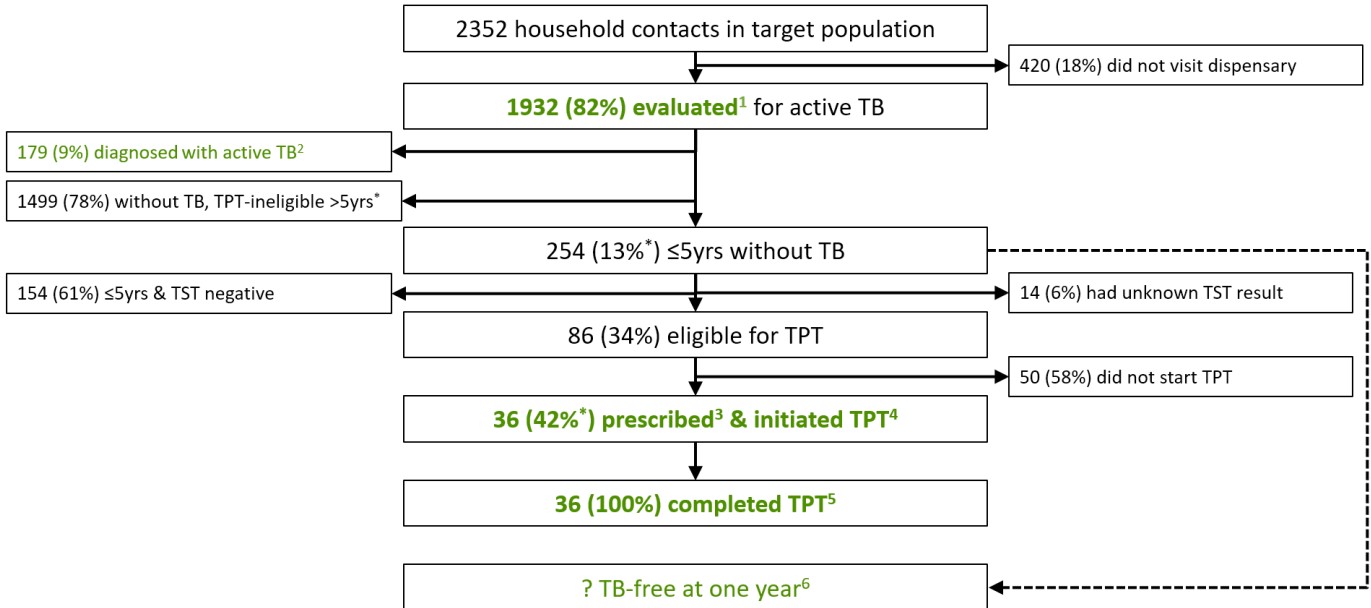

**Figure 5** Flow diagram showing the Prevent Cascade, as adapted from the Zero TB Indicator Framework for Mongolia. Flow diagrams showing individuals entering and exiting at each step of the three cascades, either appropriately to the left side after having received guideline-adherent care or inappropriately to the right side after having received guideline-non-adherent care. The adapted Zero TB Treat indicators are shown in green and numbered using superscripts as defined in table 1. The indicators in **bolded text** describe the quality of care at a given step in the cascade, while the non-bolded indicators describe the diagnostic and/or epidemiological yield of that step in TB care. *Percentages derived from multivariate, logistic regression models using generalising estimations and may differ from crude proportions. ?, unknown. TB, tuberculosis.

relationship to index patient, and the age and gender of the index patient (online supplemental table S3).

Of the household contacts who reached the dispensary, 179 (9%) were diagnosed with active TB, yielding 7600 cases per 100 000, providing a number-needed-to-screen among all household contacts of 13. Concurrently, 1753 (91%) contacts were found not to have active TB, including 254 (15%; 95% CI: 13% to 16%) under 5 years of age who were eligible for screening for LTBI. Eighty-six (34%; 95% CI: 29% to 41%) child contacts were tuberculin skin test (TST) positive. Among these, 36 (43%; 95% CI: 33% to 55%) were prescribed and initiated isoniazid preventive therapy (IPT) and all 36 (100%) completed TPT. Of all 8518 individuals entering the Prevent Cascade as household contacts being screened for active TB, 1868 (79.4%) received all services defined as indicative of high-quality care in terms of timeliness and efficiency[7 8] (figure 3C).

Older index patient age was associated with substantially lower odds of an eligible child being prescribed preventive therapy (aOR: 0.60 per decade, 95% CI: 0.38 to 0.93, p=0.022), after adjusting for age and gender of the contact, the contact's relationship to index patient and gender of the index patient (online supplemental table S4).

## DISCUSSION

We systematically evaluated the quality of TB care at the subnational and facility levels by applying the Zero TB Indicator Framework to routinely collected TB surveillance data from four administrative regions of Mongolia. Overall, the quality of care in the Search Cascade was high, with almost everyone receiving all recommended active case-finding services. In contrast, the quality of care in the Prevent Cascade was only moderate-to-high, with only four out of every five receiving all recommended household TB screening and prevention services, and the quality of care in the Treat Cascade was only moderate, with only two out of every three receiving all recommended passive case-finding and treatment services. The exercise also enabled us to localise important gaps in the efficiency, individual and public health effectiveness, and equity of TB active and passive case-finding and prevention in reference to evidence-based standards. In addition, we identified individual and facility-level factors associated with poor outcomes, providing a starting point for quality improvement interventions.

This work is an important contribution in several ways. First, it is among the first efforts to assimilate analyses of quality across the spectrum of TB services, including TB active case-finding, TB diagnosis and treatment and TPT simultaneously. Second, the cascades were constructed directly using routinely collected data, without the requirement for intermediate epidemiological assumptions, an important advance towards the goal of integrating actionable, real-time quality assessments into routine TB surveillance at local level. Third, using the Zero TB Indicator Framework to achieve this goal provided the opportunity to evaluate its feasibility as a tool for monitoring the

quality of TB care in a resource-constrained, high TB-incidence country, a critical need in global public health.

In addition, our analysis illustrates how the systematic evaluation of care processes using the Zero TB Indicator Framework could enable national programmes to identify areas for additional evaluation and quality improvement activities. First, in Mongolia, although 96% of community members invited to screen in the 'Search' cascade ultimately received all recommended services, the cumulative effectiveness as inferred from the diagnostic and treatment yield was unexpectedly low, barely exceeding the national TB prevalence rate. The low yield is partly attributable to the use of sputum smear microscopy for confirmatory testing, a low sensitivity diagnostic test in the context of active case-finding for early stage disease, rather than mycobacterial culture or molecular methods as recommended in international guidelines.[41] These higher quality diagnostics remain prohibitively costly, even in middle-income countries; continued efforts are needed to increase financial commitments to advance TB elimination efforts by implementing internationally recommended diagnostics. Also, the absolute number of participants reached and the additional cases identified by these small, short-term active case-finding initiatives were modest, pointing to the need to better target active case-finding towards TB 'hotspots'[42] and to sustain screening efforts over time.[4] In contrast, the yield of active TB case-finding among household contacts in the Prevent Cascade was an order of magnitude higher and contact investigation also facilitated TB prevention efforts, potentially offering much greater value.

Second, we identified the largest gaps in quality of care within the Treat Cascade, where small numbers of patients dropping out from individual care processes accumulated into large losses across the cascade. In Mongolia's centralised system for TB evaluation, one-third of symptomatic individuals did not reach diagnostic centres, a metric that varied widely by geographic area. Even after patients reached TB dispensaries, diagnostic yield and treatment completion also varied widely. While one low-volume, rural site delivered effective diagnosis and treatment to 80% of those presenting for evaluation, only one-quarter to one-half of patients at the other three sites completed all the Treat Cascade steps that we defined as high quality of care. In addition, treatment completion was below the national average at all four sites, especially among older adults and men, who were also less likely to initiate treatment than younger adults and women.[1] These data suggest a need to further investigate patient-level barriers to accessing diagnostics and treatment at Mongolia's centralised TB dispensaries. In a systematic review of nearly 60 studies of TB diagnostic and treatment delay worldwide, initial presentation to low-level government facilities, geographic isolation and poverty were three of the most important risk factors.[43] According to a 2017 Mongolia NTP survey, >50% patients with TB are unemployed and 70% have incomes below the poverty line. If our data are representative of the broader symptomatic

population identified at primary health centres, as found elsewhere,[44 45] travel and related opportunity costs may be prohibitive to completing referrals and treatment. Decentralising TB care to primary health centres is one policy intervention to be considered. In a small study from Southern Brazil, a variety of diagnostic and treatment outcomes were better in a municipality with decentralised care than in a comparable municipality with centralised care.[46] In central Sulawesi, Indonesia, a TB programme that began offering education, evaluation and treatment in a community-based setting tripled their case-detection rates relative to baseline and also increased treatment completion considerably.[47]

Third, in our evaluation of the Prevent Cascade, almost 80% of household contacts completed TB evaluation, approaching the WHO's 90% target.[48] This compares favourably to the 20%–53% who completed contact investigation in prior studies, and the number needed to screen of 13 was far below the 56 reported in a prior systematic review.[34 35 49 50] Nevertheless, over half of child contacts eligible for TPT did not start it, similar to results reported elsewhere,[50–52] and the likelihood of initiation decreased substantially with increasing age of index patients. In a prior systematic review,[36] a low perceived risk of active TB for one's children was among the most frequent reasons for low uptake, emphasising the need for personalised education and counselling to address parental concerns. In addition, high rates of TPT completion among a small group of children and high rates of referral completion among adults suggest additional capacity to expand TPT in Mongolia to older child and adult contacts, as recommended by the 2018 WHO guidelines.[5]

## Strengths

Our study had several strengths. Most prior studies have focused on only one specific aspect of TB services such diagnosis and treatment, or prevention. In contrast, we adopted a more comprehensive perspective,[19] evaluating multiple evidence-based interventions concurrently using 14 indicators within the Zero TB Indicator Framework. We successfully adapted the indicators to fit local data and guidelines and developed evidence-based targets as comparators. Second, building on previous studies that relied on research data, aggregated national data or incorporated epidemiologic assumptions about TB prevalence,[53] we generated cascades directly from data collected at the basic TB management units and referring diagnostic centres. We believe this approach could help address the global need to operationalise standardised measures of quality within routine data collection systems, so that cascades may be actionable at facility and district level.[13 26] Although this study's manual extraction procedures would not be feasible for routine use, our approach could help justify adapting routine TB registers to include process measures of quality. Third, we identified individual and health-system factors associated with failure to complete key steps of the cascade, providing important information for tailoring future quality improvement interventions.

## Limitations

Our study also had some limitations. First, we were unable to evaluate 1 year postcompletion outcomes of active TB treatment and TPT as called for by the Zero TB indicator Framework. Although these follow-up data are important for evaluating effectiveness, few programmes in high-incidence, resource-constrained countries collect it, and the feasibility and costs of doing so must be considered given low rates of relapse of active TB[54] or progression of latent TB after treatment completion.[5] Second, our study sites may not be representative of all sites in Mongolia, given our purposive selection strategy. However, evaluating a large random sample was beyond the scope of this preliminary evaluation, so we selected communities representing different population densities, housing types, income levels and economic settings. Additionally, the high quality of Search Cascade services may reflect the greater resources available to the implementing NGOs and therefore not be generalisable. Third, some records may have been duplicated because we relied on aggregated data for some analyses and lacked unique identifiers to link different visits to the same individual. Similarly, some outcome misclassification may be related to inaccurate tracking of handwritten records across multiple official primary data sources, and some records were incomplete. These challenges highlight the need to standardise and expand use of Mongolia's nascent electronic TB registration system (ie, TUBIS.mn), to improve both the quality of reporting and the quality of care. In Zambia, for example, regular review and cross-validation of TB registers at diagnostic and treatment units was associated with significant reductions in the proportion of non-assessed cases.[55] Finally, we did not evaluate the readiness of the systems and structures in these districts to deliver TB care, which would provide important and complementary information for assessing quality.[49]

## Conclusion

We found that using standardised treatment indicators and cascades to evaluate core processes of evidence-based TB interventions was feasible and informative for evaluating the quality of TB services. This approach may be generalisable to other high-burden settings that employ WHO's standardised approach to TB monitoring and reporting.[56 57] To complement such analyses, there is a need to incorporate qualitative and mixed-methods studies[31 58 59] to contextualise the barriers and facilitators of care processes, as well as structural assessments of quality such as service availability and readiness surveys,[60] to help enhance the outcomes of TB care.[61] Above all, it is time for researchers and practitioners to strengthen routine TB surveillance data systems to collect rigorous, accessible, and reliable data on the quality of TB services worldwide.

**Author affiliations**
¹Zero TB Mongolia, Mongolian Health Initiative, Ulaanbaatar, Mongolia
²Department of Epidemiology of Microbial Diseases, Yale School of Public Health, New Haven, Connecticut, USA
³Department of Biostatistics, Yale School of Public Health, New Haven, Connecticut, USA
⁴Center for Methods in Implementation and Prevention Science, Yale School of Public Health, New Haven, Connecticut, USA
⁵School of Public Health, Mongolian National University of Medical Sciences, Ulaanbaatar, Mongolia
⁶Ulaanbaatar City Health Department, Governor's Office of Capital City Ulaanbaatar, Ulaanbaatar, Mongolia
⁷Department of Hospital Development, Mongolian National University of Medical Sciences, Ulaanbaatar, Mongolia
⁸Tuberculosis Surveillance and Research Department, National Center for Communicable Diseases, Ulaanbaatar, Mongolia
⁹Department of Global Health and Social Medicine, Harvard Medical School, Boston, Massachusetts, USA
¹⁰Department of Nutrition, Harvard T. H. Chan School of Public Health, Boston, Massachusetts, USA
¹¹Pulmonary, Critical Care, and Sleep Medicine Section, Department of Medicine, Yale School of Medicine, New Haven, Connecticut, USA

**Acknowledgements** The authors wish to acknowledge the Mongolian National TB Program; the doctors, nurses and staff at the primary health centers and the TB dispensaries who provided the care under evaluation in this project; the patients and community members whose data contributed to this analysis; Courtney Yuen, Mercedes Becerra and Michael Wilson for their assistance in establishing the Zero TB Initiative in Mongolia; The Mongolian Association of Family Medicine Specialists, and the staff of the Mongolian Health Initiative.

**Contributors** JLD, DG and DS helped in study conceptualisation. Data curation was performed by AS, CP, KD, TS and CA. Formal analysis was done by AS, CP, XZ and JLD. Funding acquisition was obtained by DS. JLD, XZ and DS were responsible for methodology. Project administration was performed by AS, DG, JLD, GG, TD, AC and MBB. Verification of the underlying data was done by AS, KD, DG and SE. CP and JLD performed the study visualisation. AS, CP and JLD were involved in writing the original draft. Review and editing were performed by all the authors (AS, CP, XZ, KD, TS, SE, AC, TD, GG, MBB, DS, DG, JLD). DG is the guarantor.

**Funding** This work was sponsored by a grant from the National Institute of Environmental Health Sciences (DP1 ES025459 to DS) and an anonymous foundation based in the UK. Christina Parisi is funded by NIAAA T32AA025877. The study funders had no role in design, collection, analysis or interpretation of the data, in writing this report, or in the decision to submit it for publication.

**Competing interests** None declared.

**Patient and public involvement** Patients and/or the public were not involved in the design, or conduct, or reporting or dissemination plans of this research.

**Patient consent for publication** Not applicable.

**Ethics approval** Institutional review boards at Yale University (#2000025951), the Harvard School of Public Health (#19-1489) and the Mongolian National University of Medical Sciences (#2019/3-10) approved the study protocol as minimal risk research and waived requirements for informed consent.

**Provenance and peer review** Not commissioned; externally peer reviewed.

**Data availability statement** The data that support the findings of this study are available from the corresponding author, Ariunzaya Saranjav, upon reasonable request.

**ORCID iDs**
Ariunzaya Saranjav http://orcid.org/0000-0001-9122-6686
Meredith B Brooks http://orcid.org/0000-0003-1022-2515
J Lucian Davis http://orcid.org/0000-0002-8629-9992

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
