## [Reviewer comments · BMJ Open]

ARTICLE DETAILS

TITLE (PROVISIONAL)	Assessing the Quality of Tuberculosis Care Using Routine Surveillance Data: A process evaluation employing the Zero TB Indicator Framework in Mongolia
AUTHORS	Saranjav, Ariunzaya; Parisi, Christina; Zhou, Xin; Dorjnamjil, Khulan; Samdan, Tumurkhuyag; Erdenebaatar, Sumiya; Chuluun, Altantogoskhon; Dalkh, Tserendagva; Ganbaatar, Gantungalag; Brooks, Meredith; Spiegelman, Donna; Ganmaa, Davaasambu; Davis, J. Lucian

VERSION 1 – REVIEW

REVIEWER	Manuel Jorge University of the Philippines Manila, Physiology
REVIEW RETURNED	13-Mar-2022

GENERAL COMMENTS	Abstract (page 4-5)  1. Rewrite abstract in usual manner using complete sentences. Include methodology. 2. Summary box: rewrite using complete sentences. page 14 line 51  1. Were there individuals that arrived but did not complete TB testing?
---

REVIEWER	Silvia Chiang Brown University, Pediatrics
REVIEW RETURNED	15-Mar-2022

GENERAL COMMENTS	The study's aim is to evaluate the feasibility of the Zero TB Indicator Framework for assessing the quality of TB services using routinely collected data from Mongolia's NTP. The study design is innovative, and the data are important for working towards the goal of TB elimination. The figures are clear and informative. I do have significant concerns about the manuscript, but most are related to a lack of clarity in the methods section. Depending on the answers to these questions, I may have additional comments/questions, if the manuscript is revised and resubmitted. The introduction and discussion sections are clear and well written. METHODS  1. The study population and inclusion/exclusion criteria are unclear. "Search study": Who are the high-risk populations? Which NGOs? How were the patients identified? "Treat study": How were the patients identified? What are the indications to refer patients for evaluation at TB dispensaries? What symptoms do they have to have? Do they have to have positive tests of infection? Were all 3
---

	“studies” conducted at the described study sites? Why were these sites selected? Prior collaborations with the investigators? Other reasons? What were the exclusion criteria? 2. It is unusual to say that a cohort study was nested w/in a cross-sectional study; usually it’s the other way around. I would change the wording because it’s confusing. a. I recommend saying “Search cohort” or “Search cascade” instead of “Search study”; ditto for “Treat” and “Prevent.” b. Then you can just clarify which cohort was longitudinal and which were cross-sectional. 3. Table 1 a. Search: i. What criteria must be met for someone to be considered “screened for possible active TB disease”? Symptom screening + CXR? ii. What are the symptoms on the screening checklist, and what CXR findings were considered compatible w/ TB? iii. What does the TB evaluation consist of? Did the evaluation differ between study sites? iv. How was TB disease defined? Microbiological confirmation and/or clinical diagnosis? If clinical diagnoses were included, how were they made? v. How was “beginning active TB treatment” defined? Was there a minimum number of doses that one had to take? b. Treat: i. Same as Search #4. ii. Which assays were used for microbiological confirmation? Culture only? NAAT? AFB smear microscopy? iii. Same as Search #5. iv. How was treatment success defined? c. Prevent: numbering of items needs to be fixed i. TB screening – same as Search #1 ii. TB evaluation – same as Search #3 iii. TB diagnosis – same as Search #4 iv. TPT eligibility – what is considered TST positivity? Were child contacts evaluated differently than adult contacts? Was a follow-up TST administered 8-10 weeks after the index case was considered no longer infectious? Was any TPT given to the youngest patients (<2 years or <5 years) even if TST was negative in case the TST was unreliable (e.g., not enough time to convert to positive)? 4. Page 11 lines 48-55: how were the process outcomes w/ the greatest losses identified (i.e., what constitutes “greatest loss”? 5. An ethics statement is needed. If appropriate, the authors can just say that IRB approval was not needed because deidentified data were used, or something similar. RESULTS 6. Page 13 lines 28-30; page 14 lines 17-18: what is considered “high-quality care”? DISCUSSION 7. Given that NGOs were involved in the Search cascade, isn’t it possible that they devoted additional resources to patient follow-up, and that could help explain the higher patient retention as compared to the other two cascades?
--	--

REVIEWER	Shahed Hossain icddr,b, Centre for Equity and Health Systems
REVIEW RETURNED	19-Mar-2022

GENERAL COMMENTS	"Assessing the Quality of Tuberculosis Care Using Routine Surveillance Data: A process evaluation employing the Zero TB Indicator Framework in Mongolia" (Manuscript ID bmjopen-2022-061229) General Comments: This is a well-structured and focused paper reporting the feasibility of a novel tool "Zero TB indicator framework" to assess the quality of TB case finding, treatment, and prevention services in Mongolia- a Zero TB Initiative country. The authors were very succinct in laying out the need for the study, a good description of the "Zero TB indicator framework" and its adaptation in this research, the methods, the comparatively newer approach of cascade analysis (which seems to be interesting and appropriate in this case) and the results. The discussion was adequate and they have covered all the strengths and possible limitations that they have noted during the research. They have appropriately identified the service weaknesses and where the NTP needs its attention, i.e., weak referral, loss to follow up, and monitoring mechanisms and processes. I appreciate the authors providing adequate supplementary Information in the appendices and supplementary files including the systematic review search strategy. One suggestion: since the study compared among the four sites (including one hospital), it would be more appreciable to know how similar these sites were at the time of the study, not in terms of their catchment population, but in terms of their systems preparation including human resources and their training status, logistics supplies, TB diagnostics and availability of guidelines, health information systems, and monitoring mechanisms, etc. Also, if the centers were ever prepared for WHO ISTC guidelines by NTP Mongolia or not? Above all, this is well-written paper and I wish wide circulation of it. Thanks.
--

REVIEWER	Rukman Awang Hamat Univ Putra Malaysia
REVIEW RETURNED	24-Mar-2022

GENERAL COMMENTS	Overall, this is an interesting study detailing the changes in the three TB delivery cascades within the Zero TB Indicator Framework in Mongolia. Specific comments are as follows: Introduction: It would be good to include some issues or areas of concerns about Zero TB Initiative or surveillance in Mongolia. What happened to the exhibiting TB surveillance or evaluation system in Mongolia? Methods: Page 10, first paragraph-I would provide some figures for the
---

	densely populated areas in those study sites. Analysis: Page 13-I would mention p-values of less than 0.05 are considered significant. Results and discussion: Page 15-Even though the authors state earlier that they used a reference for bacteriologically confirmed, I would like to know how they confirmed these cases. Other variables such as drug-resistant TB, successfully treated TB, drug-susceptible and resistant cases, and failed TB treatment. I would suggest the authors to define these variables in the methodology section. Figure 3-about 38 (3%) dd not start treatment. Although the percentage is small but the implication if big that is the risk of transmission to susceptible individuals. I would like the authors to discuss about this concern. And what have been done for patients with failed TB treatment. Figure 5B and 5C- It would be good to replace a question mark with other symbols and try to describe it in the footnote.
--	---

VERSION 1 – AUTHOR RESPONSE

Reviewer: 1

Dr. Manuel Jorge, University of the Philippines Manila

We thank Dr. Manuel Jorge for the enthusiastic review of our research manuscript and for the discretionary comments.

Comments to the Author:

Abstract (page 4-5)

1. Rewrite abstract in usual manner using complete sentences. Include methodology.

Thank you for your comment. We have updated the Abstract to include the study design and to include complete sentences, where possible while remaining within the scope of BMJ Open guidelines on Abstracts

ABSTRACT

Objectives: To evaluate the feasibility of the Zero TB Indicator Framework as a tool for assessing the quality of tuberculosis (TB) case-finding, treatment, and prevention services in Mongolia.

Setting: Primary health centers, tuberculosis dispensaries, and surrounding communities in four districts of Mongolia.

Design: Three retrospective cross-sectional cohort studies, and two longitudinal studies, each individually nested in one of the cohort studies.

Participants: 15,947 community members from high tuberculosis-risk populations; 8,518 patients screened for tuberculosis in primary health centers and referred to dispensaries; 857 index tuberculosis patients and 2352 household contacts.

Primary and Secondary Outcome Measures: 14 indicators of the quality of tuberculosis care defined by the Zero TB Indicator Framework and organized into three care cascades, evaluating community-based active case-finding, passive case finding in health facilities, and TB screening and prevention among close contacts; individual and health-system predictors of these indicators.

Results: The cumulative proportions of participants receiving guideline-adherent care varied widely, from 96% for community-based active case-finding, to 79% for TB preventive therapy among

household contacts, to only 57% for passive case-finding in primary health centers and TB dispensaries (range 29% to 80% across districts). The odds of patients completing active TB treatment decreased substantially with increasing age (aOR 0.76 per decade, 95%CI 0.71-0.83, $p<0.001$) and among men (aOR 0.56, 95%CI 0.36-0.88, $p=0.013$). Contacts of older index patients also had lower odds of initiating and completing of TB preventive therapy (aOR 0.60 per decade, 95%CI 0.38-0.93, $p=0.022$).

Conclusions: The Zero TB Framework provided a feasible and adaptable approach for using routine surveillance data to evaluate the quality of TB care and identify associated individual and health system factors associated. Future research should evaluate strategies for collecting process indicators more efficiently; gather qualitative data on explanations for low-quality care; and deploy quality improvement interventions.

2. Summary box: rewrite using complete sentences.

We have removed the Summary box section, as requested from the journal. Thank you for your understanding.

Page 14 line 51

1. Were there individuals that arrived but did not complete TB testing?

The data source that we used to confirm arrival at the facility was the TB laboratory register; therefore, we unfortunately do not have information about those who may have arrived but failed to undergo TB testing. However, to be more precise in our language, we have adjusted our description of this step in the evaluation process:

Of the 8,518 individuals screened, 1,486 (17%) were assigned non-TB diagnoses. The remaining 7,032 (83%) were referred to dispensaries for TB evaluation, but only 4,416 (63%) arrived and underwent and completed TB testing. Those tested included 2,348 males (53%) and four people living with HIV (0.1%). Median age was 40 years (lower quartile (Q1) 26 to upper quartile(Q3) 55).

Reviewer: 2

Prof. Silvia Chiang, Brown University

Comments to the Author:

The study's aim is to evaluate the feasibility of the Zero TB Indicator Framework for assessing the quality of TB services using routinely collected data from Mongolia's NTP. The study design is innovative, and the data are important for working towards the goal of TB elimination. The figures are clear and informative. I do have significant concerns about the manuscript, but most are related to a lack of clarity in the methods section. Depending on the answers to these questions, I may have additional comments/questions, if the manuscript is revised and resubmitted. The introduction and discussion sections are clear and well written.

We thank Prof. Silvia Chiang for the enthusiastic review of our research protocol and for the discretionary comments. A primary goal of this manuscript is to demonstrate the feasibility and adaptability of a cascade framework to assess the quality of TB care as delivered in a routine manner and documented using routinely collected and recorded surveillance data in reference to national and international guidelines. Below we respond to the questions raised and provide additional detail about these national and international guidelines as requested.

My discretionary comments:

METHODS

1. The study population and inclusion/exclusion criteria are unclear. "Search study": Who are the high-risk populations? Which NGOs? How were the patients identified?

We have added further detail about the high-risk populations, NGOs and identification of patients who were eligible for TB screening, as follows:

The Search cascade examined community-based, active TB case-finding among selected high-risk populations (people living in poverty; school staff and students; household contacts of TB index patients) screened by NTP at NCCD, and its partner non-governmental organizations (e.g., Korean National Tuberculosis Association; Mongolian Health Initiative). Screened high-risk populations were identified using data from the Mongolian National Statistics Office on census tracts having lower socioeconomic status, records from TB index patients, recommendations from TB unit staff of a school with a recent TB outbreak, and records from TB index cases.

"Treat study": How were the patients identified? What are the indications to refer patients for evaluation at TB dispensaries? What symptoms do they have to have? Do they have to have positive tests of infection?

We have clarified indications for referring patients for evaluation at TB dispensaries:

The Treat cascade examined clinic-based, TB diagnosis and treatment among possible TB patients identified in primary health centers and referred to dispensaries, including first episodes of care only. The indications for referral followed NTP guidelines and included reporting one or more TB symptoms, defined as a cough and/or fever lasting ≥ 2 weeks, or weight loss or blood-tinged sputum; or being found to have abnormalities on chest radiography.

Testing for TB infection is performed at TB dispensaries.

Were all 3 "studies" conducted at the described study sites? Why were these sites selected? Prior collaborations with the investigators? Other reasons? What were the exclusion criteria?

We have also added detailed background information below about the chosen sites, all of which contributed data to each of the cascades:

Bayanzurkh (population 280,000), Khan-Uul (population 170,000), and Chingeltei (population 150,000) are three densely populated districts among nine total districts in the Mongolian capital, Ulaanbaatar, all of which have both urban and rural areas. Mandal is the highest-population sub-province in rural Selenge province. Case notification rates in all four communities approximate the national average case notification rate of 124/100,000 persons. We chose Bayanzurkh District because it has the highest TB rate in Ulaanbaatar; Khan-Uul District because it is a partner district for the Mongolia Zero TB Initiative and has established active surveillance and screening programs at high schools; and Chingeltei because it was recognized as a model district at the 2018 National TB Forum. Mandal has two prisons and three large mines, both of which are high-risk settings for TB. Bayanzurkh has 15 TB staff (7 TB specialist doctors, 6 TB nurses, 2 laboratory technicians); Khan-Uul, 7 TB staff (4 TB specialist doctors, 2 TB nurses, 1 laboratory technicians). Chingeltei, 5 TB staff (2 specialist TB doctors, 2 nurses, 1 laboratory technician); and Mandal 7 TB staff (1 TB doctor, 5 TB nurses, 1 laboratory technician). Each dispensary also has 2-4 TB volunteers.

2. It is unusual to say that a cohort study was nested w/in a cross-sectional study; usually it's the other way around. I would change the wording because it's confusing.

Thank you, we understand that this terminology is a bit confusing. As suggested, we have changed the terminology to “retrospective cross-sectional cohort” studies for the Search, Treat, and Prevent cascades. There is precedent for this terminology in the literature (see Hudson, J. I., Pope, H. G., Jr, & Glynn, R. J. (2005). The cross-sectional cohort study: an underutilized design. *Epidemiology* (Cambridge, Mass.), 16(3), 355–359. <https://doi.org/10.1097/01.ede.0000158224.50593.e3>). For the Treat and Prevent cascades, we had individual data so we conducted nested longitudinal cohort studies to understand longitudinal outcomes. We have changed the wording to make this clearer.

a. I recommend saying “Search cohort” or “Search cascade” instead of “Search study”; ditto for “Treat” and “Prevent.”

We have changed this language to Search cascade, Treat cascade, and Prevent cascade. Thank you for your recommendation.

b. Then you can just clarify which cohort was longitudinal and which were cross-sectional.

We have clarified this under the heading “Study Design, Participants, and Sites” in the Methods section:

For the Search, Treat, and Prevent cascades, we conducted retrospective cross-sectional cohort studies. For the Treat and Prevent cascades, we had access to individual data, so we were able to conduct nested longitudinal cohort studies to examine long-term outcomes of treatment in addition to the baseline cross-sectional cohort studies.

3. Table 1

We have added further details about how each indicator was defined, as footnotes to Table 1, and summarized these changes below.

a. Search:

i. What criteria must be met for someone to be considered “screened for possible active TB disease”?
Symptom screening + CXR?

bScreening includes questions about a history of TB symptoms (cough or fever lasting ≥ 2 weeks, weight loss, or bloody sputum) and/or CXR, with any symptom or CXR abnormality (e.g., consolidation, infiltrates, nodules, cavities) defining a positive screen.

ii. What are the symptoms on the screening checklist, and what CXR findings were considered compatible w/ TB?

Please see response to Reviewer 2, point 3(a)i above.

iii. What does the TB evaluation consist of? Did the evaluation differ between study sites?

cCompleting TB evaluation is defined as receiving a clinical assessment and all tests required by the TB clinician at the TB dispensary. According to NTP guidelines, these may include direct (sputum smear microscopy, mycobacterial culture, or Xpert MTB/RIF assay) or indirect (biopsy, adenosine deaminase test (ADA), TST) tests.

iv. How was TB disease defined? Microbiological confirmation and/or clinical diagnosis? If clinical diagnoses were included, how were they made?

dDiagnosed with active TB disease is defined as being recorded in the National TB Program TB lab register and/or the treatment register as a new TB patient, confirmed as of one of the following categories: bacteriologically confirmed drug-susceptible TB, bacteriologically confirmed drug-resistant TB, bacteriologically confirmed multidrug-resistant TB, bacteriologically confirmed extensively drug-resistant TB, or unconfirmed TB based on clinician judgment.

v. How was “beginning active TB treatment” defined? Was there a minimum number of doses that one had to take?

We have changed this to Prescribed active TB treatment, as defined below:

eDefined as being recorded in the National TB Program treatment register as a new TB patient prescribed TB treatment.

b. Treat:

i. Same as Search #4.

Please see response to Reviewer 2, point 3(a)iv above.

ii. Which assays were used for microbiological confirmation? Culture only? NAAT? AFB smear microscopy?

fBacteriologic confirmation may occur by sputum smear microscopy, sputum mycobacterial culture, and/or Xpert MTB/RIF.

iii. Same as Search #5.

Please see response to Reviewer 2, point 3(a)v above.

iv. How was treatment success defined?

gTreatment outcome successful was defined to include all patients who completed the recommended duration of treatment, whether documented to have negative smear microscopy or a mycobacterial culture result within the final month of treatment (“TB cured”, per NTP guidelines) or not (“Completed TB treatment”, per NTP guidelines). Patients lost to follow-up (i.e., those who did not start treatment or had treatment interrupted for ≥ 2 consecutive months); those not evaluated; and those who transferred to another treatment unit in a different area, were defined as treatment outcome not successful.

c. Prevent: numbering of items needs to be fixed

We have fixed the numbering. Thank you for your suggestion.

i. TB screening – same as Search #1

hThose eligible for screening for possible preventive TB treatment include all household and other close contacts of a TB index patient. Screening should be performed within 14 days following the diagnosis of an TB index patient.

ii. TB evaluation – same as Search #3

iAccording to NTP guidelines, contacts 0-15 years of age should be evaluated with a TST (defined as positive for TB infection among close contacts when ≥ 5 mm skin induration is documented within 2-3 days of placement); TST may be repeated 12 weeks after exposure. For contacts ≥ 15 years, a chest X-ray (CXR) is required. If TST is positive or CXR abnormalities are identified, bacteriological testing with sputum Xpert MTB/RIF or mycobacterial culture is recommended. If TB is bacteriologically confirmed, active TB treatment is recommended. For children under 5 years of age who are TST positive and in whom active TB has been excluded, TB preventive therapy is recommended, and may be prescribed in other situations (e.g., concerns about false negative results) at the clinician's discretion.

iii. TB diagnosis – same as Search #4

Please see response to Reviewer 2, point 3(a)iv above.

iv. TPT eligibility – what is considered TST positivity? Were child contacts evaluated differently than adult contacts? Was a follow-up TST administered 8-10 weeks after the index case was considered no longer infectious? Was any TPT given to the youngest patients (<2 years or <5 years) even if TST was negative in case the TST was unreliable (e.g., not enough time to convert to positive)?

Please see response to Reviewer 3, point 3(c)ii above.

4. Page 11 lines 48-55: how were the process outcomes w/ the greatest losses identified (i.e., what constitutes “greatest loss”?

We have added further information to clarify:

“Finally, for process outcomes with the greatest losses (i.e., the largest percentage drops in retention between steps), we constructed multivariate, logistic regression models using GEE to identify individual demographic (age, gender) predictors of dropout, with 95% CI and p-values provided by Wald tests.”

5. An ethics statement is needed. If appropriate, the authors can just say that IRB approval was not needed because deidentified data were used, or something similar.

Thank you. An ethics statement now appears in the second to last paragraph of the methods, as well as at the end of the manuscript, as per BMJ Open house style.

RESULTS

6. Page 13 lines 28-30; page 14 lines 17-18: what is considered “high-quality care”?

We defined high-quality care as receiving each of the services indicated in the cascade, without dropping out. We have rephrased this for greater clarity:

“Of all 15,947 community members screened for active TB in the Search Cascade, 15,261(95.7%) completed each of the required steps of the cascade, an indication of high-quality care (Figure 5a).”

DISCUSSION

7. Given that NGOs were involved in the Search cascade, isn't it possible that they devoted additional resources to patient follow-up, and that could help explain the higher patient retention as compared to the other two cascades?

Thank you for bringing up this important point. We have addressed it the Limitations section of the Discussion:

Additionally, the high quality of Search cascade services may reflect the greater resources available to the implementing NGOs and therefore not be generalizable.

Reviewer: 3

Dr. Shahed Hossain, icddr,b

Comments to the Author:

“Assessing the Quality of Tuberculosis Care Using Routine Surveillance Data: A process evaluation employing the Zero TB Indicator Framework in Mongolia”

(Manuscript ID bmjopen-2022-061229)

General Comments:

This is a well-structured and focused paper reporting the feasibility of a novel tool “Zero TB indicator framework” to assess the quality of TB case finding, treatment, and prevention services in Mongolia- a Zero TB Initiative country.

The authors were very succinct in laying out the need for the study, a good description of the “Zero TB indicator framework” and its adaptation in this research, the methods, the comparatively newer approach of cascade analysis (which seems to be interesting and appropriate in this case) and the results.

The discussion was adequate and they have covered all the strengths and possible limitations that they have noted during the research. They have appropriately identified the service weaknesses and where the NTP needs its attention, i.e., weak referral, loss to follow up, and monitoring mechanisms and processes.

I appreciate the authors providing adequate supplementary Information in the appendices and supplementary files including the systematic review search strategy. One suggestion: since the study compared among the four sites (including one hospital), it would be more appreciable to know how similar these sites were at the time of the study, not in terms of their catchment population, but in terms of their systems preparation including human resources and their training status, logistics supplies, TB diagnostics and availability of guidelines, health information systems, and monitoring mechanisms, etc. Also, if the centers were ever prepared for WHO ISTC guidelines by NTP Mongolia or not?

Above all, this is well-written paper and I wish wide circulation of it.

Thanks.

We thank Dr. Shahed Hossain for the enthusiastic review of our research protocol and for the discretionary comments. We appreciate the comments and agree that a formal readiness assessment would have enhanced our evaluation, but site-specific data of this kind at the beginning of this study in 2017 is not available to us, given the retrospective study design and our focus on using routinely collected data. Nevertheless, we have added general information on readiness in the Methods under Setting:

The NTP oversees management of the paper registries used for TB reporting, as well as the entry of data from these source documents into the national electronic TB information database. NTP compiles surveillance data quarterly and provides technical oversight and assistance to TB providers in Mongolia's 21 provinces and 9 capital city districts. Since 2017, the NTP has implemented enhanced human resource management practices for health workers in TB dispensaries, including on-the-job training in TB care, performance incentives, and retirement benefits.

Also, we have provided detailed information about the study sites and the human resources at each of the TB dispensaries in the Methods under Study Design, Participant and Sites:

Bayanzurkh (population 280,000), Khan-Uul (population 170,000), and Chingeltei (population 150,000) are three densely populated districts among nine total districts in the Mongolian capital, Ulaanbaatar, all of which have both urban and rural areas. Mandal is the highest-population sub-province in rural Selenge province. Case notification rates in all four communities approximate the national average case notification rate of 124/100,000 persons. We chose Bayanzurkh District because it has the highest TB rate in Ulaanbaatar; Khan-Uul District because it is a partner district for the Mongolia Zero TB Initiative and has established active surveillance and screening programs at high schools; and Chingeltei because it was recognized as a model district at the 2018 National TB Forum. Mandal has two prisons and three large mines, both of which are high-risk settings for TB. Bayanzurkh has 15 TB staff (7 TB specialist doctors, 6 TB nurses, 2 laboratory technicians); Khan-Uul, 7 TB staff (4 TB specialist doctors, 2 TB nurses, 1 laboratory technicians). Chingeltei, 5 TB staff (2 specialist TB doctors, 2 nurses, 1 laboratory technician); and Mandal 7 TB staff (1 TB doctor, 5 TB nurses, 1 laboratory technician). Each dispensary also has 2-4 TB volunteers.

We have added the lack of data on systems preparation as an additional limitation of the study:

Finally, we did not evaluate the readiness of the systems and structures in these districts to deliver TB care, which would provide important and complementary information for assessing quality (57).

Reviewer: 4

Dr. Rukman Awang Hamat, Univ Putra Malaysia

Comments to the Author:

Overall, this is an interesting study detailing the changes in the three TB delivery cascades within the Zero TB Indicator Framework in Mongolia.

We thank Dr. Rukman Awang Hamat for the enthusiastic review of our research protocol and for the discretionary comments.

Specific comments are as follows:

Introduction:

It would be good to include some issues or areas of concerns about Zero TB Initiative or surveillance in Mongolia. What happened to the exhibiting TB surveillance or evaluation system in Mongolia?

We have added detailed information about TB surveillance to Methods under Setting:

The NTP oversees management of the paper registries used for TB reporting, as well as the entry of data from these source documents into the national electronic TB information database. NTP compiles surveillance data quarterly and provides technical oversight and assistance to TB providers in Mongolia's 21 provinces and 9 capital city districts.

Methods:

Page 10, first paragraph-I would provide some figures for the densely populated areas in those study sites.

We have added population counts to the Methods under Study Design, Participant and Sites:

Bayanzurkh (population 280,000), Khan-Uul (population 170,000), and Chingeltei (population 150,000) are three densely populated districts among nine total districts in the Mongolian capital, Ulaanbaatar, all of which have both urban and rural areas. Mandal is the highest-population sub-province in rural Selenge province and has established active surveillance and screening programs at high schools; and Chingeltei because it was recognized as a model district at the 2018 National TB Forum. Mandal has two prisons and three large mines, both of which are high-risk settings for TB.

Analysis:

Page 13-I would mention p-values of less than 0.05 are considered significant.

We have now addressed this in the Methods. Thank you.

For t-tests and chi-square tests for significance, a p-value <0.05 was considered significant.

Results and discussion:

Page 15-Even though the authors state earlier that they used a reference for bacteriologically confirmed, I would like to know how they confirmed these cases. Other variables such as drug-resistant TB, successfully treated TB, drug-susceptible and resistant cases, and failed TB treatment. I would suggest the authors to define these variables in the methodology section.

As noted in the Methods under Setting, we extracted the numbers of TB cases from the source data, the national TB register at each dispensary. Definitions of bacteriologically confirmed and other forms of TB are based on Mongolian National Guidelines. In addition, we have provided further details about variables as footnotes to the table. Thank you for your suggestion.

dDiagnosed with active TB disease is defined as being recorded in the National TB Program TB lab register and/or the treatment register as a new TB patient, confirmed as of one of the following categories: bacteriologically confirmed drug-susceptible TB, bacteriologically confirmed drug-resistant TB, bacteriologically confirmed multidrug-resistant TB, bacteriologically confirmed extensively drug-resistant TB, or unconfirmed TB based on clinician judgment.

gTreatment outcome successful was defined to include all patients who completed the recommended duration of treatment, whether documented to have negative smear microscopy or a mycobacterial culture result within the final month of treatment ("TB cured", per NTP guidelines) or not ("Completed TB treatment", per NTP guidelines). Patients lost to follow-up (i.e., those who did not start treatment or had treatment interrupted for ≥ 2 consecutive months); those not evaluated; and those who transferred to another treatment unit in a different area, were defined as treatment outcome not successful.

Figure 3-about 38 (3%) dd not start treatment. Although the percentage is small but the implication if big that is the risk of transmission to susceptible individuals. I would like the authors to discuss about this concern. And what have been done for patients with failed TB treatment.

Thank you for your comment. We agree with your concern. As now noted in the Methods,

Patients with TB treatment failure are managed according to national guidelines, which in 2017 recommended repeat sputum examination for mycobacterial culture and drug-susceptibility testing, and empiric initiation of a WHO Category II re-treatment regimen.

Figure 5B and 5C- It would be good to replace a question mark with other symbols and try to describe it in the footnote.

Thank you for the suggestion. To improve clarity, we have added a statement to the Legend explaining that the question mark indicates that follow-up data on treatment outcomes at one year were not available as these are not routinely collected in Mongolia. We hope that this additional description addresses the Reviewer's concern.

VERSION 2 – REVIEW

REVIEWER	Silvia Chiang Brown University, Pediatrics
REVIEW RETURNED	22-Jun-2022

GENERAL COMMENTS	Thank you to the authors for their thorough and detailed responses to my previous comments. I have no further concerns about the paper and think it will make a great addition to the literature.
---

REVIEWER	Rukman Awang Hamat Univ Putra Malaysia
REVIEW RETURNED	24-Jun-2022

GENERAL COMMENTS	The authors have satisfactorily responded to the previous suggestions. Thank you. The revised draft is clearly written but I have a few minor recommendations as follows: Page 9- please remove the open bracket. ..(NGOs), I find that "p" is not written in italics on same pages. Kindly follow the author guidelines. Page 17-typographical error is observed. For example "..varied substantially(p<0.001)" Page 17-Table S0 is mentioned. I would rename it as Table S1. Please change all Tables accordingly. Kind regards.
--